# Learning attention for historical text normalization by learning to pronounce

## Abstract

Automated processing of historical texts often relies on pre-normalization to modern word forms. Training encoder-decoder architectures to solve such problems typically requires a lot of training data, which is not available for the named task. We address this problem by using several novel encoder-decoder architectures, including a multi-task learning (MTL) architecture using a grapheme-to-phoneme dictionary as auxiliary data, pushing the state-of-the-art by an absolute 2% increase in performance. We analyze the induced models across 44 different texts from Early New High German. Interestingly, we observe that, as previously conjectured, multi-task learning can learn to focus attention during decoding, in ways remarkably similar to recently proposed attention mechanisms. This, we believe, is an important step toward understanding how MTL works.

## 1 Introduction

There is a growing interest in automated processing of historical documents, as evidenced by the growing field of digital humanities and the increasing number of digitally available collections of historical documents. A common approach to deal with the high amount of variance often found in this type of data is to perform spelling normalization (Piotrowski, 2012), which is the mapping of historical spelling variants to standardized/modernized forms (e.g. *vnd → und* 'and').

Training data for supervised learning of historical text normalization is typically scarce. On the other hand, neural networks are said to work best when trained on large amounts of data. It is therefore not clear whether neural networks are a good choice for this particular task.

We explore framing the spelling normalization task as a character-based sequence-to-sequence transduction problem, and use encoder–decoder recurrent neural networks (RNNs) to induce our transduction models. This is similar to models that have been proposed for neural machine translation (e.g., Cho et al. (2014)), so essentially, our approach could also be considered a form of character-based neural machine translation.

By basing our model on individual characters as input, we keep the vocabulary size small, which in turn reduces the model's complexity and the amount of data required to train it effectively. Using an encoder–decoder architecture removes the need for an explicit character alignment between historical and modern wordforms. Furthermore, we explore using an auxiliary task for which data is more readily available, namely grapheme-to-phoneme mapping (word pronunciation), to regularize the induction of the normalization models.

We propose several architectures, including multi-task learning architectures taking advantage of the auxiliary data, and evaluate them across 44 small datasets from Early New High German.

**Contributions** Our contributions are as follows:

- We are, to the best of our knowledge, the first to propose and evaluate encoder-decoder architectures for historical text normalization.

- We evaluate several such architectures across 44 datasets of Early New High German.

- We show that such architectures benefit from bidirectional encoding, beam search, and attention.

- We also show that MTL with pronunciation as an auxiliary task improves the performance of architectures without attention.

- We analyze the above architectures and show that the MTL architecture *learns* attention from the auxiliary task, making the attention mechanism largely redundant.

- We make our implementation publicly available at `anonymized`.

In sum, we both push the state-of-the-art in historical text normalization and present an analysis that, we believe, brings us a step further in understanding the benefits of multi-task learning.

## 2 Datasets

**Normalization** For the normalization task, we use a total of 44 texts from the Anselm corpus (Dipper and Schultz-Balluff, 2013) of Early New High German.[1] The corpus is a collection of manuscripts and prints of the same core text, a religious treatise. Although the texts are semi-parallel and share some vocabulary, they were written in different time periods (between the 14th and 16th century) as well as different dialectal regions, and show quite diverse spelling characteristics. For example, the modern German word *Frau* 'woman' can be spelled as *fraw/vraw* (Me), *frawe* (N2), *frauwe* (St), *fraüwe* (B2), *frow* (Stu), *vrowe* (Ka), *vorwe* (Sa), or *vrouwe* (B), among others.[2]

All texts in the Anselm corpus are manually annotated with gold-standard normalizations following guidelines described in Krasselt et al. (2015). For our experiments, we excluded texts from the corpus that are shorter than 4,000 tokens, as well as a few for which annotations were not yet available at the time of writing (mostly Low German and Dutch versions). Nonetheless, the remaining 44 texts are still quite short for machine-learning standards, ranging from about 4,200 to 13,200 tokens, with an average length of 7,350 tokens.

For all texts, we removed tokens that consisted solely of punctuation characters. We also lowercase all characters, since it helps keep the size of the vocabulary low, and uppercasing of words is usually not very consistent in historical texts. Tokenization was not an issue for pre-processing these texts, since modern token boundaries have already been marked by the transcribers.

---

[1] https://www.linguistics.rub.de/anselm/

[2] We refer to individual texts using the same internal IDs that are found in the Anselm corpus (cf. the website).

**Grapheme-to-phoneme mappings** We use learning to pronounce as our auxiliary task. This task consists of learning mappings from sequences of graphemes to the corresponding sequences of phonemes. We use the German part of the CELEX lexical database (Baayen et al., 1995), particularly the database of phonetic transcriptions of German wordforms. The database contains a total of 365,530 wordforms with transcriptions in DISC format, which assigns one character to each distinct phonological segment (including affricates and diphthongs). For example, the word *Jungfrau* 'virgin' is represented as ′jUN-frB.

## 3 Model

### 3.1 Base model

We propose several architectures that are extensions of a base neural network architecture, closely following the sequence-to-sequence model proposed by Sutskever et al. (2014). It consists of the following:

- an embedding layer that maps one-hot input vectors to dense vectors;

- an encoder RNN that transforms the input sequence to an intermediate vector of fixed dimensionality;

- a decoder RNN whose hidden state is initialized with the intermediate vector, and which is fed the output prediction of one timestep as the input for the next one; and

- a final dense layer with a softmax activation which takes the decoder's output and generates a probability distribution over the output classes at each timestep.

For the encoder/decoder RNNs, we use long short-term memory units (LSTM) (Hochreiter and Schmidhuber, 1997). LSTMs are designed to allow recurrent networks to better learn long-term dependencies, and have proven advantageous to standard RNNs on many tasks. We found no significant advantage from stacking multiple LSTM layers for our task, so we use the simplest competitive model with only a single LSTM unit for both encoder and decoder.

By using this encoder–decoder model, we avoid the need to generate explicit alignments between the input and output sequences, which would bring up the question of how to deal with input/output

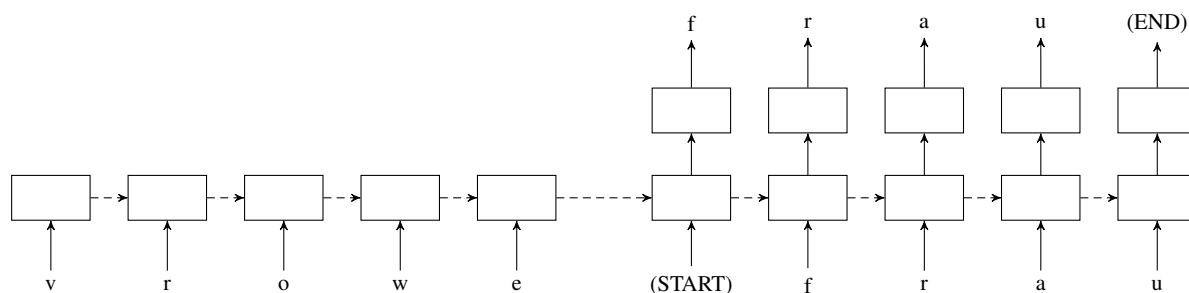

Figure 1: Flow diagram of the base model; left side is the encoder, right side the decoder, the latter of which has an additional prediction layer on top. Multi-task learning variants use two separate prediction layers for main/auxiliary tasks, while sharing the rest of the model. Embedding layers for the inputs are not explicitly shown.

pairs of different lengths. Another important property is that the model does not start to generate any output until it has seen the full input sequence, which in theory allows it to learn from any part of the input, without being restricted to fixed context windows. An example illustration of the unrolled network is shown in Fig. 1.

### 3.2 Training

During training, the encoder inputs are the historical wordforms, while the decoder inputs correspond to the correct modern target wordforms. We then train each model by minimizing the cross-entropy loss across all output characters; i.e., if $y = (y_1, ..., y_n)$ is the correct output word (as a list of one-hot vectors of output characters) and $\hat{y} = (\hat{y}_1, ..., \hat{y}_n)$ is the model's output, we minimize the mean loss $- \sum_{i=1}^{n} y_i \log \hat{y}_i$ over all training samples. For the optimization, we use the Adam algorithm (Kingma and Ba, 2015) with a learning rate of 0.003.

To reduce computational complexity, we also set a maximum word length of 14, and filter all training samples where either the input or output word is longer than 14 characters. This only affects 172 samples across the whole dataset, and is only done during training. In other words, we evaluate our models across all the test examples.

### 3.3 Decoding

For prediction, our base model generates output character sequences in a greedy fashion, selecting the character with the highest probability at each timestep. This works fairly well, but the greedy approach can yield suboptimal global picks, in which each individual character is sensibly derived from the input, but the overall word is non-

sensical. We therefore also experiment with beam search decoding, setting the beam size to 5.

Finally, we also experiment with using a lexical filter during the decoding step. Here, before picking the next 5 most likely characters during beam search, we remove all characters that would lead to a string not covered by the lexicon. This is again intended to reduce the occurrence of nonsensical outputs. For the lexicon, we use all word forms from CELEX (cf. Sec. 2) plus the target word forms from the training set.[3]

### 3.4 Attention

In our base architecture, we assume that we can decode from a single vector encoding of the input sequence. This is a strong assumption, especially with long input sequences. Attention mechanisms give us more flexibility. The idea is that instead of encoding the entire input sequence into a fixed-length vector, we allow the decoder to "attend" to different parts of the input character sequence at each time step of the output generation. Importantly, we let the model learn what to attend to based on the input sequence and what it has produced so far.

Our implementation is identical to the decoder with soft attention described by Xu et al. (2015): we calculate a context vector from the encoder's output and the decoder's hidden state, and factor this vector into the hidden state of the decoder for the next timestep. Additionally, we use a *bi-directional encoder* in this scenario, comprised of one LSTM layer that reads the input sequence normally and another LSTM layer that reads it back-

---

[3]We observe that due to this filtering, we cannot reach 2.25% of the targets in our test set, most of which are Latin word forms.

wards, and attend over the concatenated outputs of these two layers.

While a precise alignment of input and output sequences is sometimes difficult, most of the time the sequences align in a sequential order, which can be exploited by an attentional component.

### 3.5 Multi-task learning

Finally, we introduce a variant of the base architecture, with or without beam search, that does multi-task learning (Caruana, 1993). The multi-task architecture only differs from the base architecture in having two classifier functions at the outer layer, one for each of our two tasks. Our auxiliary task is to predict a sequence of phonemes as the correct pronunciation of an input sequence of graphemes. This choice is motivated by the relationship between phonology and orthography, in particular the observation that spelling variation often stems from phonological variation.

We train our multi-task learning architecture by alternating between the two tasks, sampling one instance of the auxiliary task for each training sample of the main task. We use the encoder-decoder to generate a corresponding output sequence, whether a modern word form or a pronunciation. Doing so, we suffer a loss with respect to the true output sequence and update the model parameters. The update for a sample from a specific task affects the parameters of corresponding classifier function, as well as all the parameters of the shared hidden layers.

### 3.6 Hyperparameters

We used a single manuscript (B) for manually evaluating and setting the hyperparameters. This manuscript is left out of the averages reported below. We believe that using a single manuscript for development, and using the same hyperparameters across *all* manuscripts, is more realistic, as we often do not have enough data in historical text normalization to reliably tune hyperparameters.

For the final evaluation, we set the size of the embedding and the recurrent LSTM layers to 128, applied a dropout of 0.3 to the input of each recurrent layer, and trained the model on mini-batches with 50 samples each for a total of 50 epochs (in the multi-task learning setup, mini-batches contain 50 samples of each task, and epochs are counted by the size of the training set for the main task only). All these parameters were set on the B manuscript alone.

### 3.7 Implementation

We implemented all of the models in Keras (Chollet, 2015). Any parameters not explicitly described here were left at their default values in Keras v1.0.8.

## 4 Evaluation

We split up each text into three parts, using 1,000 tokens each for a test set and a development set (that is not currently used), and the remainder of the text (between 2,000 and 11,000 tokens) for training. We then train and evaluate on each of the 43 texts (excluding the B text that was used for hyper-parameter tuning) individually.

**Baselines** We compare our architectures to several competitive baselines. Our first baseline is an averaged perceptron model trained to predict output character $n$-grams for each input character, after using Levenshtein alignment with generated segment distances (Wieling et al., 2009, Sec. 3.3) to align input and output characters. Our second baseline uses the same alignment, but trains a deep bi-LSTM sequential tagger, following Bollmann and Søgaard (2016). We evaluate this tagger using both standard and multi-task learning. Finally, we compare our model to the rule-based and Levenshtein-based algorithms provided by the Norma tool (Bollmann, 2012).[4]

### 4.1 Word accuracy

We use word-level accuracy as our evaluation metric. While we also measure character-level metrics, minor differences on character level can cause large differences in downstream applications, so we believe that perfectly matching the output sequences is more useful. Average scores across all 43 texts are presented in Table 1 (see Appendix A for individual scores).

We first see that almost all our encoder-decoder architectures perform significantly better than the four state-of-the-art baselines. All our architectures perform better than Norma and the averaged perceptron, and all the MTL architectures outperform Bollmann and Søgaard (2016).

We also see that beam search, filtering, and attention lead to cumulative gains in the context of the single-task architecture – with the best architecture outperforming the state-of-the-art by almost 3% in absolute terms.

---

[4]https://github.com/comphist/norma

|  |  | Average | Std. Dev. |
|---|---|---|---|
| **Norma** |  | 77.89% | 2.99 |
| **Averaged perceptron** |  | 75.72% | 3.19 |
| **Bi-LSTM tagger** |  | 79.91% | 2.68 |
| **MTL bi-LSTM tagger** |  | 79.56% | 2.71 |
| **Base model** | GREEDY | 78.91% | 2.85 |
|  | BEAM | 79.27% | 2.80 |
|  | BEAM+FILTER | 80.46% | 2.61 |
|  | BEAM+FILTER+ATTENTION | **82.72%** | 2.24 |
| **MTL model** | GREEDY | 80.64% | 2.45 |
|  | BEAM | 81.13% | 2.47 |
|  | BEAM+FILTER | **82.76%** | 2.22 |
|  | BEAM+FILTER+ATTENTION | 82.02% | 2.41 |

Table 1: Average word accuracy across 43 texts from the Anselm dataset, evaluated on the first 1,000 tokens of each text. Evaluation on the base encoder-decoder model (Sec. 3.1) with greedy search, beam search ($k = 5$) and/or lexical filtering (Sec. 3.3), with attentional decoder (Sec. 3.4), and the multi-task learning (MTL) model using grapheme-to-phoneme mappings (Sec. 3.5).

For our multi-task architecture, we also observe gains when we add beam search and filtering, but importantly, adding attention does not help. In fact, attention hurts the performance of our multi-task architecture quite significantly. Also note that the multi-task architecture *without* attention performs on-par with the single-task architecture *with* attention.

We hypothesize that the reason for this pattern, which is not only observed in the average scores in Table 1, but also quite consistent across the individual results in Appendix A, is that our multi-task learning already learns how to focus attention.

This is the hypothesis that we will try to validate in Sec. 5: *That multi-task learning can induce strategies for focusing attention comparable to attention strategies for recurrent neural networks.*

**Sample predictions** A small selection of predictions from our models is shown in Table 2. They serve to illustrate the effects of the various settings; e.g., the base model with greedy search tends to produce more nonsense words (*ters*, *üns-get*) than the others. Using a lexical filter helps the most in this regard: the base model with filtering correctly normalizes *ergieng* to *erging* '(he) fared', while decoding without a filter produces the non-word *erbiggen*. Even for *herczenlichen* (modern *herzlichen* 'heartfelt'), where no model finds the correct target form, only the model with filtering produces a somewhat reasonable alternative (*herzgeliebtes* 'heartily loved').

In some cases (such as *gewarnet* 'warned'), only the models with attention or multi-task learning produce the correct normalization, but even when they are wrong, they often agree on the prediction (e.g. *dicke*, *herzel*). We will investigate this property further in Sec. 5.

### 4.2 Learned vector representations

To gain further insights into our model, we created t-SNE projections (Maaten and Hinton, 2008) of vector representations learned on the M4 text.

Fig. 2 shows the learned character embeddings. In the representations from the base model (Fig. 2a), characters that are often normalized to the same target character are indeed grouped closely together: e.g., historical <v> and <u> (and, to a smaller extent, <f>) are often used interchangeably in the M4 text. Note the wide separation of <n> and <m>, which is a feature of M4 that does not hold true for all of the texts, as these do not always display a clear distinction between nasals. On the other hand, the MTL model shows a better generalization of the training data (Fig. 2b): here, <u> is grouped closer to other vowel characters and far away from <v>/<f>. Also, <n> and <m> are now in close proximity.

We can also visualize the internal word representations that are produced by the encoder (Fig. 3). Here, we chose words that demonstrate the interchangeable use of <u> and <v>. Histor-

| Input | Target | Base model | | | | MTL model |
|---|---|---|---|---|---|---|
| | | GREEDY | BEAM | B+F | B+F+ATT | B+F |
| ergieng | erging | erbiggen | erbiggen | erging | erging | erging |
| herczenlichen | herzlichen | herrgelichen | herzgelichen | herzgeliebtes | herzel | herzel |
| tewr | teuer | ters | terter | terme | teurer | der |
| iüngst | jüngst | ünsget | pingst | fingst | fingst | jüngst |
| gewarnet | gewarnt | prandet | prandert | pranget | gewarnt | gewarnt |
| dick | oft | oft | oft | oft | dicke | dicke |

Table 2: Selected predictions from some of our models on the M4 text; B = BEAM, F = FILTER, ATT = ATTENTION.

ical *vnd, vns, vmb* become modern *und, uns, um*, changing the *<v>* to *<u>*. However, the representation of *vmb* learned by the base model is closer to forms like *von, vor, uor*, all starting with *<v>* in the target normalization. In the MTL model, however, these examples are indeed clustered together.

## 5 Analysis: Multi-task learning helps focus attention

Table 1 shows that models which employ *either* an attention mechanism *or* multi-task learning obtain similar improvements in word accuracy. However, we observe a decline in word accuracy for models that *combine* multi-task learning with attention.

A possible interpretation of this counter-intuitive pattern might be that attention and MTL, to some degree, learn similar functions of the input data, a conjecture by Caruana (1998). We put this hypothesis to the test by closely investigating properties of the individual models below.

### 5.1 Model parameters

First, we are interested in the weight parameters of the final layer that transforms the decoder output to class probabilities. We consider these parameters for our standard encoder-decoder model and compare them to the weights that are learned by the attention and multi-task models, respectively.[5]

We observe that the weight differences between the standard and the attention model correlate with the differences between the standard and multi-task model by a Pearson's $r$ of 0.346 (averaged across datasets, with a standard deviation of 0.315). Figure 4 illustrates these highly parallel weight changes for the different models when trained on the N4 dataset.

[5]For the multi-task models, this analysis disregards those dimensions that do not correspond to classes in the main task.

### 5.2 Final output

Next, we compare the effect that employing either an attention mechanism or multi-task learning has on the actual output of our system. We find that out of the 210.9 word errors that the base model produces on average across all test sets (comprising 1,000 tokens each), attention resolves 47.7, while multi-task learning resolves an average of 45.4 errors. Crucially, the errors that both of these approaches resolve independently of each other amount to 27.7 on average.

Attention and multi-task also introduce new errors compared to the base model (26.6 and 29.5 per test set, respectively), and again we can observe a relatively high agreement of the models (11.8 word errors are introduced by both models).

Finally, the attention and multi-task models display a word-level agreement of $\kappa$=0.834 (Fleiss).

### 5.3 Saliency analysis

Our last analysis regards the saliency of the input timesteps with respect to the predictions of our models. We follow Li et al. (2016) in calculating first-derivative saliency for given input/output pairs and compare the scores from the different models. The higher the saliency of an input timestep, the more important it is in determining the model's prediction at a given output timestep. Therefore, if two models produce similar saliency matrices for a given input/output pair, they have learned to focus on similar parts of the input during the prediction. Our hypothesis is that the attentional and the multi-task learning model should be more similar in terms of saliency scores than either of them compared to the base model.

Figure 5 shows a plot of the saliency matrices generated from the word pair *czeychen – zeichen* 'sign'. Here, the scores for the attentional and

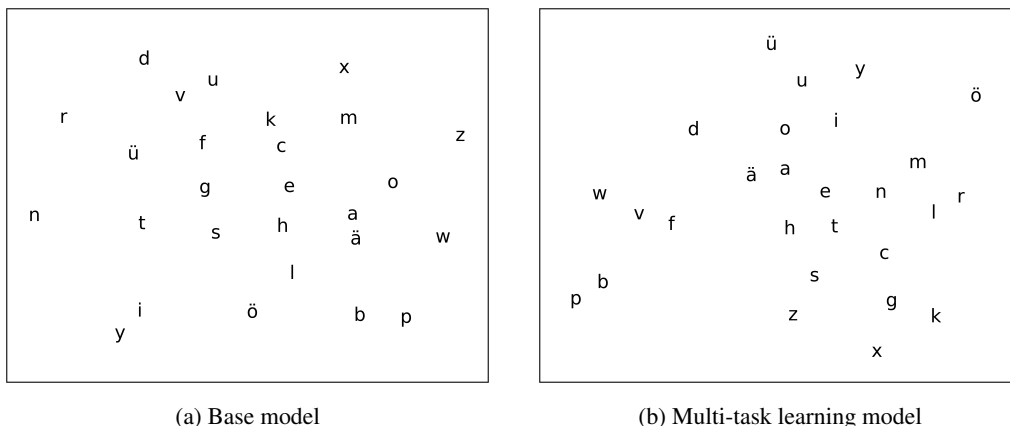

(a) Base model

(b) Multi-task learning model

Figure 2: t-SNE projections (with perplexity 7) of character embeddings from models trained on M4

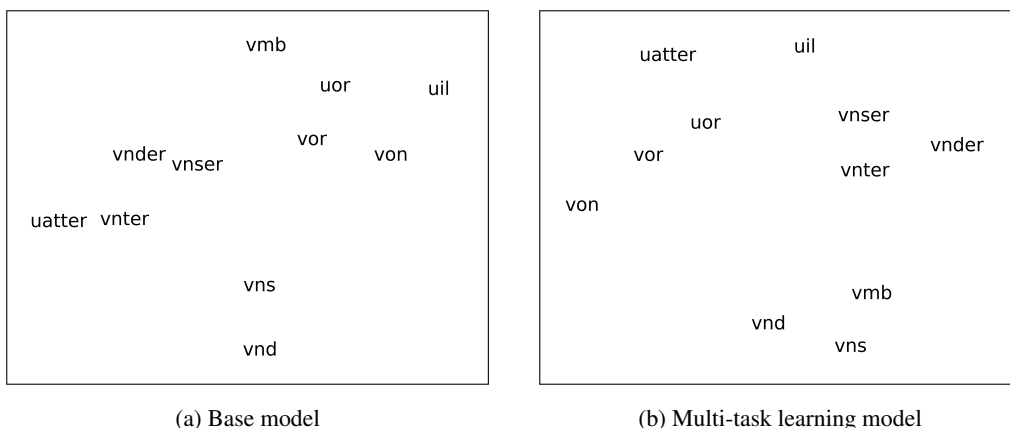

(a) Base model

(b) Multi-task learning model

Figure 3: t-SNE projections (with perplexity 5) of the intermediate vectors produced by the encoder ("historical word embeddings"), from models trained on M4

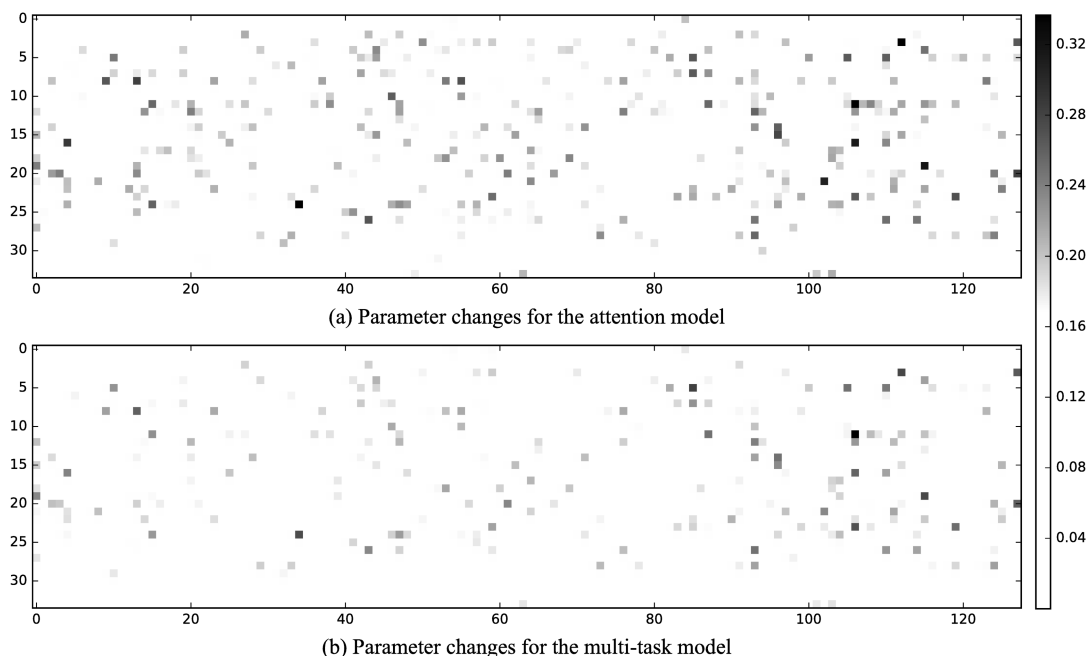

(a) Parameter changes for the attention model

(b) Parameter changes for the multi-task model

Figure 4: Heat map of parameter differences in the final dense layer between (a) the plain and the attention model as well as (b) the plain and the multi-task model, when trained on the N4 manuscript. The changes correlate by $\rho = 0.959$.

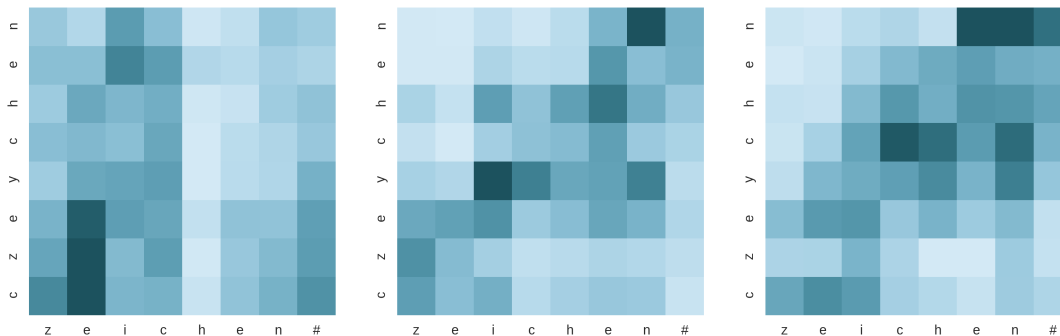

Figure 5: First-derivative saliency w.r.t. the input sequence, as calculated from the base model (left), the attentional model (center), and the MTL model (right). The scores for the attentional and the multi-task model correlate by $\rho = 0.615$, while the correlation of either one with the base model is $|\rho| < 0.12$.

the MTL model indeed correlate by $\rho = 0.615$, while those for the base model do not correlate with either of them. A systematic analysis across 19,000 word pairs (where all models agree on the output) shows that this effect only holds for longer input sequences ($\geq 7$ characters), with a mean $\rho = 0.303$ ($\pm 0.177$) for attentional vs. MTL model, while the base model correlates with either of them by $\rho < 0.21$.

## 6  Related Work

Many traditional approaches to spelling normalization of historical texts use edit distances or some form of character-level rewrite rules, hand-crafted (Baron and Rayson, 2008) or learned automatically (Bollmann, 2013; Porta et al., 2013).

A more recent approach is based on character-based statistical machine translation applied to historical text (Pettersson et al., 2013; Sánchez-Martínez et al., 2013; Scherrer and Erjavec, 2013; Ljubešić et al., 2016) or dialectal data (Scherrer and Ljubešić, 2016). This is conceptually very similar to our approach, except that we substitute the classical SMT algorithms for neural networks. Indeed, our models can be seen as a form of character-based neural MT (Cho et al., 2014).

Neural networks have rarely been applied to historical spelling normalization so far; one example is Azawi et al. (2013), who use bi-directional LSTMs, along with a layer that performs alignment between input and output wordforms, to normalize old Bible text.

## 7  Conclusion and Future Work

We presented an approach to historical spelling normalization using neural networks with an

encoder-decoder architecture, and showed that it consistently outperforms several existing baselines. Encouragingly, our work proves to be fully competitive with the sequence-labeling approach by Bollmann and Søgaard (2016), without requiring a prior character alignment.

Specifically, we demonstrated the aptitude of multi-task learning to mitigate the shortage of training data for the named task. We included a multifaceted analysis of the effects that MTL introduces to our models and the resemblance that it bears to attention mechanisms. We believe that this analysis is a valuable contribution to the understanding of MTL approaches also beyond spelling normalization, and we are confident that our observations will stimulate further research into the relationship between MTL and attention.

Finally, many improvements to the presented approach are conceivable, most notably introducing some form of token context to the model. Currently, we only consider word forms in isolation, which is problematic for ambiguous cases (such as *jn*, which can normalize to *in* 'in' or *ihn* 'him') and conceivably makes the task harder for others. Reranking the predictions with a language model could be one possible way to improve on this. Ljubešić et al. (2016), for example, experiment with segment-based normalization, using segments (essentially, token $n$-grams) instead of single tokens as the input for their character-based SMT model, which also introduces context. Such an approach could also deal with the issue of tokenization differences between the historical and the modern text, which is another challenge often found in datasets of historical text.

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

## A    Supplementary Material

For interested parties, we provide our full evaluation results for each single text in our dataset. Table 3 shows token counts, a rough classification of each text's dialectal region, and the results for the baseline methods. Table 4 presents the full results for our encoder-decoder models.

| ID | Region | Tokens | Norma | Avg. Perc. | Bi-LSTM Tagger | |
|----|--------|--------|-------|------------|----------------|---|
| | | | | | BASE | MTL |
| B | East Central | 4,718 | 79.60% | 76.30% | 79.20% | 78.82% |
| D3 | East Central | 5,704 | 79.70% | 77.20% | 80.10% | 81.62% |
| H | East Central | 8,427 | 83.00% | 78.60% | 85.00% | 84.32% |
| B2 | West Central | 9,145 | 76.20% | 74.60% | 82.00% | 80.12% |
| KÄ1492 | West Central | 7,332 | 78.40% | 74.80% | 81.60% | 80.82% |
| KJ1499 | West Central | 7,330 | 77.00% | 73.50% | 84.50% | 80.22% |
| N1500 | West Central | 7,272 | 77.60% | 72.70% | 79.00% | 78.52% |
| N1509 | West Central | 7,418 | 78.40% | 74.30% | 80.80% | 80.02% |
| N1514 | West Central | 7,412 | 78.50% | 72.20% | 79.00% | 79.62% |
| St | West Central | 7,407 | 73.30% | 70.30% | 75.50% | 73.03% |
| D4 | Upper/Central | 5,806 | 76.10% | 72.40% | 76.50% | 76.62% |
| N4 | Upper | 8,593 | 79.30% | 80.00% | 81.80% | 82.52% |
| s1496/97 | Upper | 5,840 | 81.20% | 77.70% | 83.00% | 82.62% |
| B3 | East Upper | 6,222 | 82.30% | 79.50% | 81.50% | 83.02% |
| Hk | East Upper | 8,690 | 79.10% | 78.20% | 80.90% | 79.52% |
| M | East Upper | 8,700 | 75.20% | 72.80% | 83.90% | 82.72% |
| M2 | East Upper | 8,729 | 76.30% | 75.10% | 76.70% | 79.32% |
| M3 | East Upper | 7,929 | 79.20% | 77.30% | 80.40% | 81.52% |
| M5 | East Upper | 4,705 | 81.60% | 76.40% | 77.70% | 76.92% |
| M6 | East Upper | 4,632 | 74.90% | 73.70% | 75.20% | 75.72% |
| M9 | East Upper | 4,739 | 81.00% | 79.00% | 80.40% | 79.32% |
| M10 | East Upper | 4,379 | 77.20% | 76.00% | 75.10% | 75.92% |
| Me | East Upper | 4,560 | 80.20% | 76.90% | 80.30% | 79.12% |
| Sb | East Upper | 7,218 | 79.60% | 75.70% | 80.00% | 80.12% |
| T | East Upper | 8,678 | 76.00% | 73.40% | 75.80% | 73.43% |
| W | East Upper | 8,217 | 77.60% | 78.20% | 81.40% | 80.72% |
| We | East Upper | 6,661 | 82.70% | 78.60% | 81.50% | 82.22% |
| Ba | North Upper | 5,934 | 79.10% | 80.20% | 80.70% | 80.02% |
| Ba2 | North Upper | 5,953 | 80.70% | 78.10% | 82.50% | 82.12% |
| M4 | North Upper | 8,574 | 76.70% | 75.70% | 79.40% | 79.32% |
| M7 | North Upper | 4,638 | 78.60% | 75.60% | 78.20% | 77.42% |
| M8 | North Upper | 8,275 | 79.30% | 78.20% | 81.10% | 80.02% |
| n | North Upper | 9,191 | 79.80% | 81.90% | 84.40% | 84.62% |
| N | North Upper | 13,285 | 74.00% | 71.70% | 79.00% | 79.42% |
| N2 | North Upper | 7,058 | 82.80% | 80.30% | 84.30% | 81.72% |
| N3 | North Upper | 4,192 | 78.10% | 76.40% | 77.60% | 77.12% |
| Be | West Upper | 8,203 | 74.90% | 75.30% | 78.80% | 77.52% |
| Ka | West Upper | 12,641 | 72.80% | 75.40% | 80.10% | 81.62% |
| SG | West Upper | 7,838 | 79.70% | 78.00% | 81.70% | 81.12% |
| Sa | West Upper | 8,668 | 71.50% | 71.90% | 76.10% | 74.93% |
| Sa2 | West Upper | 8,834 | 77.60% | 73.50% | 79.50% | 79.72% |
| St2 | West Upper | 8,686 | 72.80% | 73.20% | 78.20% | 79.92% |
| Stu | West Upper | 8,011 | 78.00% | 76.50% | 79.40% | 79.62% |
| Le | Dutch | 7,087 | 71.30% | 65.00% | 75.60% | 75.12% |
| *Average (-B)* | | 7,353 | 77.89% | 76.30% | 79.91% | 79.56% |

Table 3: Word accuracy on the Anselm dataset, evaluated on the first 1,000 tokens, using the baseline models (cf. Sec. 4): the Norma tool (Bollmann, 2012), an averaged perceptron model, and a deep bi-LSTM sequential tagger (Bollmann and Søgaard, 2016).

| ID | Base model | | | | Multi-task learning model | | | |
|---|---|---|---|---|---|---|---|---|
| | G | B | B+F | B+F+A | G | B | B+F | B+F+A |
| B | 76.90% | 77.30% | 78.40% | **82.70%** | 77.70% | 79.50% | 81.70% | 80.10% |
| D3 | 81.50% | 81.60% | 82.70% | **83.20%** | 81.10% | 81.70% | 82.90% | **83.20%** |
| H | 82.60% | 82.90% | 84.50% | **87.40%** | 85.00% | 85.80% | 86.60% | 85.20% |
| B2 | 81.00% | 81.20% | 82.40% | **83.40%** | 80.00% | 80.40% | 82.70% | 83.00% |
| KÄ1492 | 83.00% | 83.40% | 83.60% | 84.00% | 83.40% | 83.70% | **85.10%** | 84.90% |
| KJ1499 | 81.30% | 81.30% | 82.00% | **84.60%** | 84.00% | 84.00% | 83.80% | 82.50% |
| N1500 | 79.50% | 80.30% | 81.30% | **84.00%** | 82.20% | 82.50% | 83.60% | 82.30% |
| N1509 | 82.10% | 82.40% | 83.10% | **85.00%** | 82.80% | 83.50% | 84.50% | 82.80% |
| N1514 | 80.40% | 80.50% | 81.10% | 83.40% | 82.30% | 82.80% | **84.20%** | 83.10% |
| St | 74.60% | 74.60% | 76.40% | 79.70% | 77.60% | 77.80% | **80.20%** | 77.70% |
| D4 | 77.90% | 77.20% | 79.00% | 81.40% | 77.00% | 77.90% | **81.50%** | 79.90% |
| N4 | 82.10% | 82.30% | 82.90% | **84.80%** | 83.10% | 83.00% | 84.40% | 84.00% |
| s1496/97 | 80.40% | 80.10% | 81.10% | 82.10% | 82.30% | 82.50% | **85.20%** | 83.90% |
| B3 | 80.80% | 81.20% | 82.20% | **85.20%** | 82.70% | 83.30% | 84.80% | 84.50% |
| Hk | 77.30% | 79.00% | 79.40% | 82.90% | 80.30% | 80.40% | 81.20% | **83.70%** |
| M | 81.40% | 81.50% | 82.60% | **85.00%** | 82.90% | 82.90% | 82.70% | 84.00% |
| M2 | 79.90% | 80.50% | 81.30% | 81.80% | 78.80% | 77.80% | 79.60% | **83.20%** |
| M3 | 81.00% | 81.10% | 82.00% | **83.70%** | 82.80% | 82.50% | 83.50% | 81.70% |
| M5 | 76.60% | 77.10% | 79.00% | **82.00%** | 78.20% | 78.20% | 80.90% | 81.50% |
| M6 | 72.70% | 73.80% | 75.20% | 80.20% | 77.30% | 79.00% | **80.30%** | 76.60% |
| M9 | 78.20% | 78.50% | 79.70% | **83.20%** | 80.70% | 79.70% | **83.20%** | 79.60% |
| M10 | 72.00% | 72.40% | 73.20% | 77.40% | 75.70% | 76.30% | **77.90%** | 77.80% |
| Me | 76.90% | 76.50% | 78.50% | **81.30%** | 77.30% | 79.20% | 81.00% | 77.40% |
| Sb | 78.80% | 79.10% | 81.30% | 81.40% | 80.60% | 81.00% | **84.00%** | 82.90% |
| T | 75.60% | 75.10% | 77.40% | **80.30%** | 76.90% | 78.00% | 80.10% | 79.50% |
| W | 80.80% | 81.20% | 82.40% | 81.90% | 80.40% | 81.60% | **84.40%** | **84.40%** |
| We | 77.70% | 80.00% | 81.80% | **84.40%** | 83.00% | 82.70% | 83.80% | 83.30% |
| Ba | 81.00% | 80.60% | 80.90% | **84.00%** | 80.40% | 81.00% | 82.60% | 81.60% |
| Ba2 | 79.70% | 80.90% | 82.00% | 84.00% | 82.60% | 83.30% | **85.40%** | 85.10% |
| M4 | 78.40% | 78.60% | 79.90% | 81.00% | 82.10% | 82.20% | **82.60%** | 80.50% |
| M7 | 74.70% | 76.30% | 78.60% | 82.00% | 79.60% | 79.90% | **82.30%** | 81.10% |
| M8 | 80.80% | 81.30% | 82.50% | **85.70%** | 82.00% | 82.50% | 84.00% | 85.40% |
| n | 83.40% | 83.40% | 84.30% | 86.00% | 84.90% | 86.30% | **88.00%** | 85.50% |
| N | 77.40% | 77.40% | 79.40% | 79.80% | 80.00% | 80.30% | **81.50%** | 80.30% |
| N2 | 82.00% | 82.30% | 83.80% | 86.40% | 82.40% | 83.50% | **86.60%** | 85.80% |
| N3 | 73.60% | 74.00% | 75.10% | **81.20%** | 76.00% | 76.30% | 80.30% | 78.70% |
| Be | 75.50% | 75.40% | 77.60% | 78.10% | 78.10% | 78.40% | 79.70% | **80.20%** |
| Ka | 81.20% | 81.20% | 81.80% | **83.90%** | 81.20% | 83.10% | 83.40% | 82.30% |
| SG | 81.10% | 81.90% | 83.40% | **85.50%** | 82.60% | 84.30% | 84.90% | 83.00% |
| Sa | 76.80% | 77.20% | 78.10% | **80.60%** | 77.50% | 78.00% | 79.70% | 79.90% |
| Sa2 | 78.90% | 79.70% | 80.70% | 81.30% | 79.70% | 81.00% | **82.30%** | **82.30%** |
| St2 | 77.70% | 78.10% | 79.00% | **81.60%** | 79.60% | 79.70% | 80.50% | 80.60% |
| Stu | 77.40% | 77.30% | 78.30% | 82.50% | 82.00% | 81.80% | **83.10%** | 82.90% |
| Le | 77.40% | 78.10% | 78.20% | 79.60% | 78.30% | 78.60% | **79.80%** | 78.90% |
| *Average (-B)* | 78.91% | 79.27% | 80.46% | **82.72%** | 80.64% | 81.13% | **82.76%** | 82.02% |

Table 4: Word accuracy on the Anselm dataset, evaluated on the first 1,000 tokens, using our base encoder-decoder model (Sec. 3) and the multi-task model. G = greedy decoding, B = beam-search decoding (with beam size 5), F = lexical filter, A = attentional model. Best results (also taking into account the baseline results from Table 3) shown in bold.

