# Peer review of "Learning attention for historical text normalization by learning to pronounce"

_ACL 2017 — decision unknown_

[Official Review · Reviewer 1 · rating 4 · confidence 4]
soundness 5 · originality 5 · clarity 2 · impact 3 · substance 3 · appropriateness 5 · meaningful comparison 3 · presentation format Oral Presentation

[update after reading author response: the alignment of the hidden units does
not match with my intuition and experience, but I'm willing to believe I'm
wrong in this case.  Discussing the alignment in the paper is important (and
maybe just sanity-checking that the alignment goes away if you initialize with
a different seed).  If what you're saying about how the new model is very
different but only a little better performing -- a 10% error reduction -- then
I wonder about an ensemble of the new model and the old one.  Seems like
ensembling would provide a nice boost if the failures across models are
distinct, right?  Anyhow this is a solid paper and I appreciate the author
response, I raise my review score to a 4.]

- Strengths:

  1)  Evidence of the attention-MTL connection is interesting

  2)  Methods are appropriate, models perform well relative to state-of-the-art

- Weaknesses:

  1)  Critical detail is not provided in the paper

  2)  Models are not particularly novel

- General Discussion:

This paper presents a new method for historical text normalization.  The model
performs well, but the primary contribution of the paper ends up being a
hypothesis that attention mechanisms in the task can be learned via multi-task
learning, where the auxiliary task is a pronunciation task.  This connection
between attention and MTL is interesting.

There are two major areas for improvement in this paper.  The first is that we
are given almost no explanation as to why the pronunciation task would somehow
require an attention mechanism similar to that used for the normalization task.
 Why the two tasks (normalization and pronunciation) are related is mentioned
in the paper: spelling variation often stems from variation in pronunciation. 
But why would doing MTL on both tasks result in an implicit attention mechanism
(and in fact, one that is then only hampered by the inclusion of an explicit
attention mechanism?).                    This remains a mystery.  The paper can
leave some
questions unanswered, but at least a suggestion of an answer to this one would
strengthen the paper.

The other concern is clarity.  While the writing in this paper is clear, a
number of details are omitted.                    The most important one is the
description
of
the attention mechanism itself.  Given the central role that method plays, it
should be described in detail in the paper rather than referring to previous
work.  I did not understand the paragraph about this in Sec 3.4.

Other questions included why you can compare the output vectors of two models
(Figure 4), while the output dimensions are the same I don't understand why the
hidden layer dimensions of two models would ever be comparable.  Usually how
the hidden states are "organized" is completely different for every model, at
the very least it is permuted.                    So I really did not understand
Figure 4.

The Kappa statistic for attention vs. MTL needs to be compared to the same
statistic for each of those models vs. the base model.

At the end of Sec 5, is that row < 0.21 an upper bound across all data sets?

Lastly, the paper's analysis (Sec 5) seems to imply that the attention and MTL
approaches make large changes to the model (comparing e.g. Fig 5) but the
experimental improvements in accuracy for either model are quite small (2%),
which seems like a bit of a contradiction.

[Official Review · Reviewer 2 · rating 3 · confidence 4]
soundness 5 · originality 5 · clarity 5 · impact 3 · substance 3 · appropriateness 5 · meaningful comparison 3 · presentation format Poster

Summary:

The paper applies a sequence to sequence (seq2seq) approach for German
historical text normalization, and showed that using a grapheme-to-phoneme
generation as an auxiliary task in a multi-task learning (MTL) seq2seq
framework improves performance. The authors argue that the MTL approach
replaces the need for an attention menchanism, showing experimentally that the
attention mechanism harms the MTL performance. The authors also tried to show
statistical correlation between the weights of an MTL normalizer and an
attention-based one.

Strengths:

1) Novel application of seq2seq to historical text correction, although it has
been applied recently to sentence grammatical error identification [1]. 

2) Showed that using grapheme-to-phoneme as an auxiliary task in a MTL setting
improves text normalization accuracy.

Weaknesses:

1) Instead of arguing that the MTL approach replaces the attention mechanism, I
think the authors should investigate why attention did not work on MTL, and
perhaps modify the attention mechanism so that it would not harm performance.

2) I think the authors should reference past seq2seq MTL work, such as [2] and
[3]. The MTL work in [2] also worked on non-attention seq2seq models.

3) This paper only tested on one German historical text data set of 44
documents. It would be interesting if the authors can evaluate the same
approach in another language or data set.

References:

[1] Allen Schmaltz, Yoon Kim, Alexander M. Rush, and Stuart Shieber. 2016.
Sentence-level grammatical error identification as sequence-to-sequence
correction. In Proceedings of the 11th Workshop on Innovative Use of NLP for
Building Educational Applications.

[2] Minh-Thang Luong, Ilya Sutskever, Quoc V. Le, Oriol Vinyals, and Lukasz
Kaiser. Multi-task Sequence to Sequence Learning. ICLR’16. 

[3] Dong, Daxiang, Wu, Hua, He, Wei, Yu, Dianhai, and Wang, Haifeng. 
Multi-task learning for multiple language translation. ACL'15

---------------------------
Here is my reply to the authors' rebuttal:

I am keeping my review score of 3, which means I do not object to accepting the
paper. However, I am not raising my score for 2 reasons:

* the authors did not respond to my questions about other papers on seq2seq
MTL, which also avoided using attention mechanism. So in terms of novelty, the
main novelty lies in applying it to text normalization.

* it is always easier to show something (i.e. attention in seq2seq MTL) is not
working, but the value would lie in finding out why it fails and changing the
attention mechanism so that it works.

[Official Review · Reviewer 3 · rating 4 · confidence 3]
soundness 5 · originality 5 · clarity 5 · impact 3 · substance 5 · appropriateness 5 · meaningful comparison 3 · presentation format Oral Presentation

- Strengths: well written, solid experimental setup and intriguing qualitative
analysis

- Weaknesses: except for the qualitative analysis, the paper may belong better
to the applications area, since the models are not particularly new but the
application itself is most of its novelty

- General Discussion: This paper presents a "sequence-to-sequence" model with
attention mechanisms and an auxiliary phonetic prediction task to tackle
historical text normalization. None of the used models or techniques are new by
themselves, but they seem to have never been used in this problem before,
showing and improvement over the state-of-the-art. 

Most of the paper seem like a better fit for the applications track, except for
the final analysis where the authors link attention with multi-task learning,
claiming that the two produce similar effects. The hypothesis is intriguing,
and it's supported with a wealth of evidence, at least for the presented task. 
I do have some questions on this analysis though:

1) In Section 5.1, aren't you assuming that the hidden layer spaces of the two
models are aligned? Is it safe to do so?

2) Section 5.2, I don't get what you mean by the errors that each of the models
resolve independently of each other. This is like symmetric-difference? That
is, if we combine the two models these errors are not resolved anymore?

On a different vein, 3) Why is there no comparison with Azawi's model?

========

After reading the author's response.

I'm feeling more concerned than I was before about your claims of alignment in
the hidden space of the two models. If accepted, I would strongly encourage the
authors to make clear
in the paper the discussion you have shared with us for why you think that
alignment holds in practice.